# The Effect of Kidney Transplantation and Immunosuppressive Therapy on Adipose Tissue Content and Adipocytokine Plasma Concentration—Preliminary Study

**DOI:** 10.3390/cimb47040255

**Published:** 2025-04-07

**Authors:** Daria Śleboda-Taront, Joanna Stępniewska, Barbara Dołęgowska, Emilia Marchelek, Katarzyna Dołęgowska, Małgorzata Marchelek-Myśliwiec

**Affiliations:** 1Department of Laboratory Diagnostics, University Clinical Hospital No. 2, Powstancow Wielkopolskich 72, 70-111 Szczecin, Poland; d.sleboda12@gmail.com; 2Department of Nephrology, Transplantology and Internal Medicine, Pomeranian Medical University, Powstancow Wielkopolskich 72, 70-111 Szczecin, Poland; malgorzata.marchelek@gmail.com; 3Department of Microbiology, Immunology and Laboratory Medicine, Pomeranian Medical University, Powstancow Wielkopolskich 72, 70-111 Szczecin, Poland; barbara.dolegowska@pum.edu.pl (B.D.); kasia.dolegowska@gmail.com (K.D.); 4American University of the Caribbean School of Medicine, Sizer St., Preston PR1 1JQ, UK; marchelekemilia@gmail.com

**Keywords:** chronic kidney disease, kidney transplantation, adipocytokines, adipose tissue, immunosuppressive therapy

## Abstract

Kidney transplantation is the preferred treatment for chronic kidney disease, significantly improving patient survival and quality of life. After the procedure, there is a gradual tendency to normalize most of the physiological and metabolic processes, but the need for immunosuppression may lead to new disorders related to the drugs’ side effects and changes in proportions of body composition. The aim of the study was to analyze the concentrations of adipocytokines such as leptin, adiponectin, visfatin, and resistin, and to assess the body composition in patients with stabilized kidney graft function treated with tacrolimus, mycophenolate mofetil, and glucocorticosteroids. A total of 47 participants were enrolled, including 25 kidney transplant recipients on uniform immunosuppressive therapy and 22 healthy controls. The concentrations of leptin, adiponectin, and IL-6 in kidney transplant recipients was significantly higher than in the control group (*p* = 0.014, *p* = 0.031, *p* = 0.000, respectively), while the other adipocytokines, such as visfatin and resistin, do not obtain statistically significant differences. The bioelectrical impedance analysis showed statistically significant differences for fat-free mass index (*p* = 0.027), visceral fat area (*p* = 0.023), waist circumference (*p* = 0.006), fat mass (*p* = 0.028), and fat mass index (*p* = 0.034), all of which had higher mean values in the study group. Preliminary findings suggest that kidney transplantation leads to significant alterations in adipocytokines levels, with potential implications for metabolic health.

## 1. Introduction

Adipose tissue is a type of connective tissue that constitutes 20% to 28% of the body mass in healthy individuals. It is primarily composed of adipocytes (fat cells), as well as preadipocytes, fibroblasts, stromal cells, and macrophages suspended in the extracellular matrix, surrounded by a network of collagen fibers, as well as blood and lymphatic vessels [1,2].

Adipose tissue serves not only as an energy reservoir but also can secrete biologically active molecules known as adipocytokines. These molecules influence energy homeostasis, glucose metabolism, tissue insulin sensitivity, lipid metabolism, appetite control, immunity, as well as the regulation of angiogenesis and the functioning of the cardiovascular system (Figure 1). Adipocytokines are hormonally active proteins with opposing pro-inflammatory or anti-inflammatory actions. The pro-inflammatory effects are notably exerted by leptin (Le), resistin (R), and visfatin (V), while adiponectin is recognized for its anti-inflammatory properties [3,4].

Studies by N. Martinez-Sanchez et al. have shown that Le correlates positively with the amount of adipose tissue and is considered a potential marker associated with atherosclerosis, diabetic retinopathy, and cardiovascular autonomic neuropathy (CAN) [5,6]. Hyperleptinemia was found to develop as a compensatory mechanism to overcoming Le resistance [7]. Elevated leptin levels are present in chronic kidney disease (CKD), independent of body mass index (BMI), age, glucose, and HDL cholesterol concentrations [8,9]. After kidney transplantation (KTx), Le dynamics may be influenced by improvements in kidney function, immunosuppressive therapy, and changes in body composition. Persistent hyperleptinemia may occur in patients with suboptimal graft function and obesity [10]. The action of glucocorticoids (GKSs) and calcineurin inhibitors (CNIs) as well as increased appetite caused by improved general health leads to weight gain and contribute to insulin resistance and new-onset diabetes after kidney transplantation (NODAT) [11,12]. Elevated leptin concentration is associated with endothelial dysfunction, hypertension, and atherosclerosis, which increases the cardiovascular risk [13]. Leptin also may influence graft function by promoting inflammation and fibrosis and could serve as a biomarker of chronic allograft dysfunction or progressive graft injury [14,15].

Adiponectin enhances tissue sensitivity to insulin, inhibits gluconeogenesis, and accelerates the oxidation of fatty acids and glucose uptake [16,17]. In CKD, elevated adiponectin levels in serum are associated with increased mortality and a higher risk of anemia, left ventricular hypertrophy, impaired bone mineralization, and, paradoxically, with an increased risk of insulin resistance and generalized inflammation [18,19,20]. After Ktx adiponectin concentration tend to normalize or decrease [21]. It may protect kidney graft against fibrosis and inflammation. Elevated levels could reflect a compensatory response to ongoing injury or systemic inflammation. Low adiponectin concentration, on the other hand, may be associated with increased metabolic and cardiovascular risk and worse kidney graft outcome, although this relationship is still under investigation [11,12]. Mota-Zamorano S. et al. showed that leptin and adipocytokine gene variants influence body fat distributions and affect the incidence of complications after Ktx [22].

Visfatin is a protein that has enzymatic and hormonal functions, primarily in energy metabolism and the regulation of inflammatory processes. Visfatin is a protein secreted by adipose tissue and its levels correlate with the fat tissue amount. However, it has been found to be secreted by many other cell types, including immune cells [23].

Visfatin is involved in carbohydrate metabolism. It promotes lipogenesis and glucose uptake by adipocytes and myocytes, while inhibiting glucose release from hepatocytes. By regulating the secretion of pro-inflammatory cytokines IL-6 and TNF-α, visfatin decreases tissue sensitivity to insulin, leading to insulin resistance [24,25,26,27].

Visfatin is an important biomarker and potential factor influencing the condition of the transplanted kidney. Its concentrations in this group of patients may be related to the functioning of the transplanted organ. Its increased amount may indicate activation of the immune system, which may increase the risk of graft rejection. Increased visfatin may be a marker of chronic rejection associated with fibrosis and vascular damage [28].

Resistin plays a role in modulating inflammatory, immunological, and autoimmune responses. A positive correlation has been observed between resistin levels and inflammatory cytokines (IL-6) as well as CRP. While a negative correlation has been observed between resistin levels and estimated glomerular filtration rate (eGFR) in patients with CKD [29,30]. Resistin concentrations decrease as kidney function improves after transplantation.

In kidney transplant recipients, the amount of adipose tissue increases due to the use of immunosuppressive drugs and improvement in health. Average weight gain after kidney transplantation ranged from 10% to 35% of baseline body weight, with the greatest increase observed during the first 12 months [31].

IL-6 is a pro-inflammatory factor involved in the progression of CKD. IL-6 could be used to assess the risk of atherosclerotic cardiovascular disease (ASCVD) and cardiovascular-related mortality in CKD patients [32]. Increased IL-6 concentrations are associated with both acute and chronic kidney allograft rejection. It promotes inflammation, fibrosis, and vascular injury within the transplanted kidney. It also plays a role in antibody-mediated rejection (ABMR) by stimulating B-cell activation and antibody production. Chronic inflammation occurs as interstitial fibrosis and tubular atrophy of the kidney graft [14,15].

The aim of the study was to analyze the concentrations of adipocytokines including leptin, adiponectin, visfatin, and resistin, and to assess the body composition in patients with stabilized kidney graft function treated with a TAC + MMF + GCS regimen.

## 2. Materials and Methods

### 2.1. Characteristics of the Study and Control Groups

A total of 47 participants were enrolled in the study. The sample size was calculated based on an expected difference in mean leptin concentrations between kidney transplant recipients and healthy controls. Drawing on previously published data by Fonseca et al. and Sukackiene et al., we assumed a mean difference of approximately 8 ng/mL with a standard deviation of 10 ng/mL [33,34]. Using a significance level of α = 0.05 and a power of 1–β = 0.8 (80%), the calculation (performed for both the *t*-test for independent samples and the non-parametric Mann–Whitney U test, considering possible deviations from normality) indicated that a minimum of 21 participants per group was required. The study group consisted of 25 kidney transplant recipients treated with the following immunosuppressive therapy regimen: tacrolimus (TAC) + mycophenolate mofetil (MMF) + glucocorticoids (GCS). Another qualifying criterion was the time after Ktx between 36 and 60 months. The control group consisted of 22 healthy volunteers.

Exclusion criteria for the study group included: immunosuppressive therapy regimens other than TAC + MMF + GCS, active cancer, treatment for acute kidney transplant rejection, diabetes, active inflammatory conditions, statin use, and dietary supplement intake.

Exclusion criteria for the control group: diagnosis of any chronic disease, dietary supplement intake.

Anthropometric data of the study and control group are presented in Table 1.

All patients and healthy volunteers provided informed consent prior to participation in the study. The study was approved by the Bioethics Committee of the Pomeranian Medical University (approval number KB−0012/10/17).

### 2.2. Tested Material

Peripheral blood from volunteers was collected in the antecubital vein in the morning and after fasting (in tubes with a K2EDTA to obtain a plasma fraction and in tubes with a clotting activator to obtain a serum fraction). For the tests, peripheral blood was collected in the antecubital vein (in tubes with a K2EDTA to obtain a plasma fraction and in tubes with a clotting activator to obtain a serum fraction). Then, the blood collected on K2EDTA and serum were centrifuged. K2EDTA plasma was transferred to new tubes and kept at −80 °C until assayed. Serum was used to perform several basic biochemicals (glucose, creatinine, total cholesterol, triglycerides, LDL cholesterol, HDL cholesterol, C-reactive protein (CRP), and insulin). Body composition was assessed using the Seca mBCA 525 medical body composition analyzer (Seca). The analyzer utilizes bioelectrical impedance measurements to calculate parameters such as fat-free mass (FFM), energy expenditure and reserves, metabolic activity, and the patient’s hydration status. The HOMA -IR (Homeostatic Model Assessment of Insulin Resistance) was calculated using the formula:$$HOMA-IR=\frac{Fasting\;Insulin\;(\mu IU/mL)\times Fasting\;Glucose\;(mg/dL)}{405}$$

#### 2.2.1. Assay Procedures

Reagents and examined plasma were equilibrated to room temperature before analysis. Plasma leptin, adiponectin, resistin, and visfatin concentrations were measured in all collected samples using the enzyme-linked immunosorbent assay reagent kits from ELK Biotechnology (Table 2).

All procedures were conducted in duplicate according to the manufacturer’s instructions. Reaction products were measured using an EnVision microplate reader (Perkin Elmer, Waltham, MA, USA) at 450 nm and their concentration was calculated based on standard curves created based on the standard solutions of specific concentrations included in the kit. Laboratory parameters such as glucose, creatinine, total cholesterol, triglycerides, LDL cholesterol, and HDL cholesterol were determined in serum using the spectrophotometric method and the Cobas 6000 module c501 biochemistry analyzer from Roche (Roche, Basel, Switzerland). C-reactive protein (CRP) was determined by immunoturbidimetric assay. Insulin concentration was determined by immunoelectrochemiluminescent assay in serum on the Cobas 6000 module c501 biochemistry analyzer from Roche (Roche, Basel, Switzerland).

#### 2.2.2. Statistical Analysis

Statistical evaluation was performed using the Statistica 13.3 PL software for Windows. The distribution of variables was assessed using the Shapiro–Wilk test. To present the variables, the number of patients, range of values (minimum–maximum), median (Me), first and third quartile values (Q1–Q3), arithmetic mean, and standard deviation (SD) were reported according to their distributions. The Mann–Whitney U test for non-parametric data and the Student’s *t*-test for parametric data were used to assess differences in the studied parameters between the groups. The correlation between the parameters was analyzed using Pearson’s test or Spearman’s rank correlation test. The significance level was set at *p* < 0.05.

## 3. Results

### 3.1. The Biochemical and Immunochemical Parameters

In the between-group analysis, statistically significance was found between the mean concentrations of glucose (*p =* 0.018), triglycerides (*p =* 0.005), creatinine (*p =* 0.000), CRP (*p =* 0.000), and the HOMA-IR (*p =* 0.003), with higher mean values found in the study group. Statistically significant differences were also observed for insulin concentration (*p =* 0.013) and GFR (*p =* 0.000), with higher mean values found in the control group. No statistically significant differences were observed for any of the other biochemical parameters (Table 3).

### 3.2. The Adipocytokine Concentrations

There were no statistically significant differences between the study and control groups regarding the mean concentrations of resistin (*p =* 0.370) and visfatin (*p =* 0.693). However, statistically significant differences were found for leptin (*p =* 0.014), adiponectin (*p =* 0.031), and IL-6 (*p =* 0.000), with higher mean values found in the study group (Table 4).

### 3.3. The Bioelectrical Impedance

The bioelectrical impedance analysis showed statistically significant differences for FFMI (*p =* 0.027), VAT (*p =* 0.023), WC (*p =* 0.006), FM (*p =* 0.028), and FMI (*p =* 0.034), all of which had higher mean values in the study group. Statistically significant differences were also observed for resistance (*p =* 0.023) and reactance (*p =* 0.031), with higher mean values observed in the control group (Table 5).

### 3.4. Spearman’s Correlations

Statistically significant correlations (*p* < 0.05) based on the Spearman’s test for both the study and control groups are shown in Table 6. In the study group, adiponectin concentration was positively correlated to the insulin concentration (*p =* 0.015) and the resistin concentration was positively correlated to the total cholesterol concentration (*p =* 0.020). In the control group, IL-6 concentration was positively correlated to the total cholesterol (*p =* 0.040), triglycerides (*p =* 0.020), non-HDL cholesterol (*p =* 0.025), the leptin concentration, and the total fat content (*p =* 0.027). A statistically significant negative correlation was found between the resistin and visfatin concentrations (*p =* 0.003), as well as between the visfatin concentration and the total fat content (*p =* 0.026).

The Spearman correlation coefficients between the concentrations of respective adipocytokines and fat content parameters within the study and control group are presented in Table 7 and Table 8. No other statistically significant correlations were observed between any other listed parameters (*p* > 0.05).

## 4. Discussion

The immunosuppressive treatment regimen following a kidney transplant typically consists of three medications: glucocorticoids, calcineurin inhibitors, and mycophenolate mofetil. In the first few months, the metabolic state and renal function are not stable, thus the adipocytokine levels should be interpreted with caution [28]. Unlike the previous studies, we analyzed the concentrations of adipocytokines—leptin, adiponectin, visfatin, and resistin—in metabolically stable patients with stabilized graft kidney function.

Kidney function has modulating effects on leptin levels, because leptin is filtered by the glomerulus and degraded in the proximal tubule by megalin. The results of previous studies indicate that after kidney transplantation (KTx), leptin levels decrease, when compared to levels in hemodialysis patients [35]. In our study, we observed significantly higher mean leptin levels in the study group, in comparison to the control group, as well as a positive correlation between leptin levels and body fat content, which is consistent with the previous research findings [36]. In patients who have undergone vascularized organ transplantation, weight gain is commonly observed, partly due to the use of glucocorticoids. These drugs stimulate the hypothalamus, leading to increased appetite and caloric intake in a dose-dependent fashion [37]. However, Kokot et al. demonstrated no association between leptin levels and glucocorticoid use during a 12-month observation period. The strongest correlation was found between leptin and BMI [38]. The role of leptin and the consequences of hyperleptinemia in patients after KTx are not yet well understood; however, it could without a doubt have prognostic significance, serving as a marker reflecting the risk of acute rejection and delayed graft function [10,33]. Studies conducted in other transplant patient groups, including heart, liver, and kidney recipients, have shown that BMI, gender, cortisol, and insulin are significant independent determinants of serum leptin levels in these patients [39].

In our study, the adiponectin levels were higher in the study group. Ahsan et al. confirmed a significant decrease in adiponectin levels after kidney transplantation compared to dialysis patients. However, compared to the control group, adiponectin levels remained elevated [40]. Observational studies confirm that within the first 12 months after KTx, adiponectin levels declined, which may suggest an important role of the kidneys in the biodegradation and elimination of adiponectin [41]. A study conducted in kidney transplant recipients demonstrated that several factors determine adiponectin (ADPN) concentration before and after KTx, including kidney function, insulin resistance, BMI, and the use of immunosuppressive therapy [42]. According to Sahin et al., cyclosporine A treatment was associated with higher adiponectin levels, while lower levels were observed in patients treated with tacrolimus [43].

In the study by Serwin et al., in which the post-transplantation observation period was 12 months, no significant correlation was found between adiponectin and GFR, and no correlation between adiponectin and HOMA-IR index was confirmed at any post-transplant time point [44].

Our study showed higher levels of IL-6 in the peripheral blood of the study group when compared to the control group. In CKD, increased IL-6 concentrations were connected with uremic toxins [45]. Based on research by Cueto-Manzano et al., it was shown that after a kidney transplantation, IL-6 levels in the serum increase significantly [46]. Similar conclusions were reached in studies conducted by H. Omrani et al. [47]. In a publication by Miller et al., IL-6 was described as a player in the development of acute and chronic kidney transplant rejections [48]. In this study, a positive correlation was found between IL-6 levels and levels of triglycerides (TG), total cholesterol, and non-HDL cholesterol when looking at the control group. Zang Hanue Xu suggests that the increase in IL-6 levels is associated with dyslipidemia [49].

In contrast to the majority of earlier publications, this study did not find any statistically significant differences in resistin levels between the control and study groups. Other studies, however, indicate significantly elevated resistin levels in KTx. K. Nagy et al. suggest that higher resistin levels are associated with a greater risk of transplant organ loss [50]. In this study, a positive correlation was found between resistin levels and total cholesterol levels when analyzing the study group. Resistin promotes lipid accumulation in macrophages, which is associated with increased expressions of CD36 and the scavenger receptor SR-A, both at the protein and mRNA levels. In the work of Kunjathoor et al., it was shown that in mice lacking these receptors, LDL uptake and degradation were reduced by 75–90% [51]. Furthermore, in this study, a negative correlation between resistin and visfatin levels was observed in the control group. This relationship has not been described previously and may require further analysis, considering the makeup of the studied group along with other additional factors.

In this study, no statistically significant differences in visfatin levels were found between study and control. Pathways by which visfatin is metabolized and eliminated are still not fully understood and remain under ongoing investigations. The data on visfatin levels in peripheral blood during the course of CKD remain ambiguous. The results of previous and up-to-date studies require long-term observations.

In studies conducted by Nagy et al., higher visfatin levels were observed in kidney transplant patients compared to the control group. In the same study it was stated that visfatin is associated with inflammatory markers, and its levels positively correlate with VCAM, high-sensitivity CRP, and creatinine levels [28]. Studies on visfatin levels after kidney transplantation have also been conducted by Fadel et al. in a pediatric population. The study showed that children after KTx have lower serum visfatin levels when compared to hemodialysis patients [52]. In this study, a negative correlation was found between visfatin levels and fat mass (FM expressed in [kg]) in the control group.

Based on the research by Nicholson et al., it was shown that visfatin levels are elevated in obese individuals, because visfatin is produced by visceral adipose tissue cells [53]. The role of visfatin in kidney transplant recipients remains unclear. Patients after KTx represent a special group due to the immunosuppressive treatment. The ambiguous results of previous and up-to-date studies require long-term observations.

In the conducted study, increase in the total fat mass (FM) and abdominal fat tissue (VAT), measured by bioelectrical impedance, as well as increase in the WC index, were observed in patients Ktx in comparison to healthy individuals.

In kidney transplant recipients during the post-transplantation period, weight gain is commonly observed and affects up to 50% of patients [54]. Compared with RTx recipients without MS, patients with MS were associated with significantly higher serum leptin levels and similar adiponectin, resistin, and visfatin levels [55].

Can adipocytokines have therapeutic functions in the area of transplantology? Adiponectin has an effect on T and B lymphocytes, which may be the basis for conducting studies in the area of immunosuppression [56]. Scientists are trying to answer this question. A study conducted by Casillas-Ramírez et al. on animals showed that the administration of adiponectin and resistin, but not visfatin, prevented fatty liver before organ harvesting by activating the protein kinase pathway: adenosine-monophosphate-activated protein kinase AMPK and phosphatidylinositol 3-kinase-Akt (PI3K-AKT) [57]. In the protective pathway, AMPK-adiponectin-resistin-PI3K/Akt may be one of the key elements of the strategy for obtaining livers from donors with extended criteria [58]. Can adipocytokines serve as markers in the art of transplantation? Studies conducted among the heart transplant patients have shown a correlation between the serum visfatin concentrations and incidents of cardiac allograft vasculopathy (CAV) [59]. Within the case-control subset, it was shown that the liver transplant patients’ pre-transplant serum adiponectin levels and hs-CRP were associated with an increased risk of cardiovascular diseases [60]. In the lung transplant patients, levels of visfatin decreased within the second month of a successful transplantation. These findings suggest that visfatin might have a proinflammatory role in end-stage lung disease (ESLD) [61]. Glucocorticoids are an essential element in transplant treatments. They elicit their immunosuppressive effects by decreasing T-lymphocytes activation via decreasing expression of interleukins IL-1, IL-2, IL-3 along with TNF-α (Tumor Necrosis Factor-α) and IFN-γ (Interferon-γ) [62]. In addition, they can be used in the treatment of acute T-cell mediated organ rejection [63]. Exogenous glucocorticoids have an influence on adipocytokines. They increase transcription of Neuropeptide Y and promote resistance to leptin [64]. In addition, it has been shown that in patients after kidney transplantation, the use of glucocorticosteroids is correlated with higher adiponectin levels [65].

However, no effect of exogenous glucocorticosteroids on resistin levels has been proven [66,67]. Our preliminary study demonstrated that in stable patients after a kidney transplantation, adipocytokine levels were not associated with the metabolic parameters.

### Study Limitations

The presented study has some worth-mentioning limitations, including the determination of adipocytokine concentrations only at a single time point and a small sample size. This is because we adopted inclusion criteria for the study, which included a uniform immunosuppression regimen based on TAC+MMF+GCS, no metabolic disorders in the form of diabetes, stable function of the transplanted kidney, and no previous episodes of acute rejection. For the analysis of body composition, we chose the bioelectrical impedance analysis (BIA) method due to its ease of performance and low cost compared to the reference methods: computed tomography, magnetic resonance imaging, and dual-energy X-ray absorptiometry. Bioelectrical impedance analysis (BIA) has undergone significant technological advances and is commonly used in scientific research [68]. Factors that impact hydration, such as recent exercise, dehydration, or food consumption, can inadvertently affect the results. In order to eliminate potential measurement errors, patients were examined fasting, in the morning, had not previously performed physical exercise, and did not present clinical signs of dehydration.

## 5. Conclusions

The adipocytokine profile, adipose tissue composition, basic parameters of carbohydrate and lipid metabolism, and correlations found in the study group indicate metabolic disorders present in patients after kidney transplantation compared to healthy individuals. The cause of the disorders is multifactorial. The study shows preliminary findings that require confirmation by research on a larger cohort, considering demographic, anthropometric, medical, and lifestyle data both before and after kidney transplantation. It could be crucial to compare the level of adipocytokines before transplantation and throughout a long-term observation up to a few years. The complex effects of adipocytokines can be challenging to interpret, especially in the organ transplant recipient population.

## Figures and Tables

**Figure 1 cimb-47-00255-f001:**
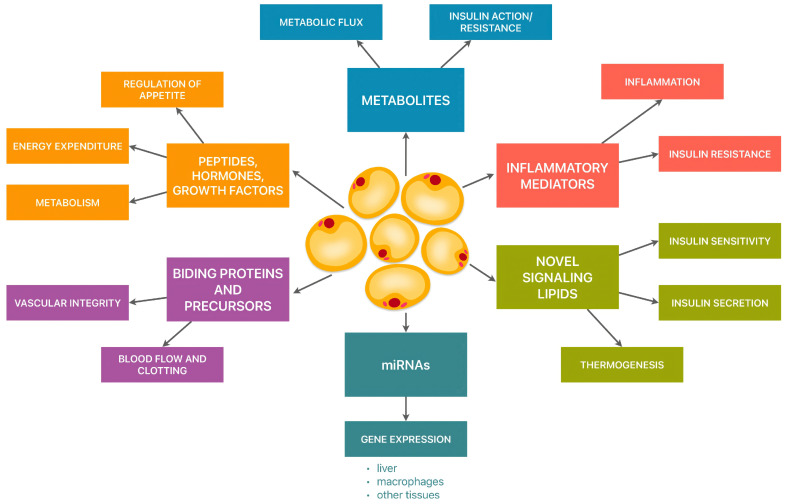
The role of adipose tissue. Endocrine functions of adipose tissue—mechanism of action. Adipose tissue serves as a key endocrine organ, actively regulating the body’s metabolic and hormonal homeostasis. The presented scheme illustrates the diverse functions of adipocytes, including the secretion of bioactive substances that influence metabolism, immunity, insulin regulation, and the cardiovascular system.

**Table 1 cimb-47-00255-t001:** Anthropometric data of the study and control group.

**Parameter**	**Study Group (B)** **(n = 25, M/F = 14/9)**				**Control Group (K)** **(n = 22, M/F = 7/15)**				** *p* **
	**Mean ± SD**	**Me** **LQ, UQ**	**Min**	**Max**	**Mean ± SD**	**Me** **LQ UQ**	**Min**	**Max**	
Age [years]	49.43 ± 10.88	51.00 43.00 57.00	26.00	64.00	37.00 ± 11.20	33.50 27.25 44.75	23.00	58.00	0.0005
Height [cm]	170.09 ± 10.69	170.00 162.00 180.50	150.00	188.00	169.64 ± 8.39	168.00 165.00 173.75	155.00	189.00	0.88
Weight [kg]	82.13 ± 16.58	90.00 67.60 94.75	49.50	108.50	69.99 ± 15.66	64.13 58.03 78.50	52.60	113.00	0.02
Waist circumference [cm]	95.22 ± 12.87	94.00 89.00 103.00	68.00	125.00	80.09 ± 11.14	80.00 70.00 87.25	68.00	105.00	0.0001
Hip circumference [cm]	103.78 ± 8.91	101.00 99.50 104.00	92.00	125.00	99.09 ± 8.45	98.00 93.00 103.75	87.00	125.00	0.08
WHR	0.92 ± 0.09	0.91 0.86 1.00	0.73	1.08	0.81 ± 0.07	0.81 0.75 0.83	0.73	1.02	0.00003
BMI [kg/m^2^]	28.11 ± 3.98	26.88 25.87 31.25	20.61	35.40	24.21 ± 4.91	22.52 20.97 26.21	18.45	39.80	0.005

BMI—body mass index; LQ—lower quartile; UQ—upper quartile; SD—standard deviation; Me—median; Min—minimum; Max—maximum; *p*-level of significance; WHR—waist to hip ratio; M/F = male/female.

**Table 2 cimb-47-00255-t002:** ELISA kit data summary.

Type	Detection Range	Sensitivity	Intra-Assay Precision	Inter-Assay Precision
Human Leptin ELISA Kit	0.16–10 ng/mL	0.06 ng/mL	CV% < 8%	CV% < 10%
Human Adiponectin ELISA Kit	0.47–30 ng/mL	75 pg/mL	CV% < 8%	CV% < 10%
Human Resistin ELISA Kit	0.16–10 ng/mL	0.058 ng/mL	CV% < 8%	CV% < 10%
Human Visfatin ELISA Kit	0.16–10 ng/mL	0.059 ng/mL	CV% < 8%	CV% < 10%

Type

**Table 3 cimb-47-00255-t003:** Statistical parameters of biochemical and immunochemical assays in the study and control groups.

Parameter	Study Group (SG)				Control lk				*p*
	Mean ± SD	Me LQ, UQ	Min	Max	Mean ± OS	Me LQ, UQ	Min	Max	
Glucose [mg/dL]	99.50 ± 12.94	99.00 96.00 102.00	72.00	141.00	93.82 ± 10.75	93.30 88.6 99.1	76.30	129.50	0.018
Insulin [µU/mL]	5.41 ± 5.20	3.15 1.67 7.63	0.34	19.78	8.38 ± 5.65	7.55 5.02 10.00	2.00	30.00	0.013
HOMA-IR	2.83 ± 1.22	2.63 2.13 3.45	0.98	6.55	1.93 ± 1.16	1.71 1.22 2.45	0.44	5.91	0.003
Total cholesterol [mg/dL]	210.76 ± 72.09	182.00 171.00 242.00	146.00	483.00	184.50 ± 25.09	184.00 166.00 200.00	140.00	247.00	0.388
HDL [mg/dL]	61.08 ± 23.93	52.00 45.00 77.00	22.00	113.00	59.91 ± 17.54	62.00 43.00 71.00	30.00	102.00	0.890
LDL [mg/dL]	135.68 ± 70.14	114.00 94.00 168.00	71.00	407.00	118.86 ± 27.74	113.00 99.00 131.00	68.00	176.00	0.765
TG [mg/dL]	135.48 ± 56.60	148.00 85.00 166.00	47.00	286.00	89.14 ± 38.50	85.50 64.00 95.00	31.00	196.00	0.005
Non-HDL cholesterol [mg/dL]	149.52 ± 67.84	139.00 115.00 167.00	71.00	414.00	124.36 ± 28.06	124.00 102.00 139.00	84.00	180.00	0.166
Creatinine [mg/dL]	1.49 ± 0.65	1.50 1.02 1.74	0.69	3.28	0.84 ± 0.12	0.80 0.77 0.92	0.64	1.06	0.000
eGFR [mL/min/1.73]	58.88 ± 26.72	54.00 40.00 81.00	16.00	116.00	100.00 ± 13.65	103.00 89.00 107.00	72.00	121.00	0.000
CRP [mg/l]	11.65 ± 4.54	11.10 8.09 14.10	4.40	19.00	1.75 ± 2.56	1.12 0.39 1.25	0.12	11.16	0.000

SD—standard deviation; Me—median; LQ—lower quartile; UQ—upper quartile; Min—minimum; Max—maximum; HOMA-IR—Homeostatic Model Assessment of Insulin Resistance; HDL—high-density lipoprotein; LDL—low-density lipoprotein; TG—triglycerides; *p*—significance level; GFR—glomerular filtration rate; CRP—C-reactive protein.

**Table 4 cimb-47-00255-t004:** Statistical parameters of adipocytokine concentrations in the study and control groups.

Parameter	Study Group (SG)				Control Group (C)				*p*
	Mean ± SD	Me LQ, UQ	Min	Max	Mean ± SD	Me LQ, UQ	Min	Max	
Leptin [ng/mL]	16.16 ± 24.83	7.40 3.54 10.74	0.11	91.80	3.86 ± 2.93	3.25 1.96 4.66	0.52	12.87	0.014
Adiponectin [ng/mL]	2.66 ± 1.89	2.39 1.45 3.17	0.32	8.63	1.62 ± 0.81	1.71 1.04 1.94	0.31	3.10	0.031
Resistin [ng/mL]	0.41 ± 0.22	0.38 0.28 0.51	0.03	0.96	0.54 ± 0.38	0.43 0.27 0.81	0.02	1.49	0.370
Visfatin [ng/mL]	1.94 ± 0.62	2.04 1.71 2.34	0.38	2.82	1.99 ± 0.63	2.19 1.74 2.42	0.71	2.89	0.693
IL6 [pg/mL]	6.19 ± 5.34	4.60 2.90 6.10	1.40	23.50	2.94 ± 4.71	1.40 1.40 2.30	1.40	22.90	0.000

SD—standard deviation; Me—median; LQ—lower quartile; UQ—upper quartile; Min—minimum; Max—maximum; *p*—significance level, the analysis was performed using a U Mann–Whitney test.

**Table 5 cimb-47-00255-t005:** Statistical parameters of measurements determined by bioelectrical impedance in the study (SG) and control (C) groups.

Parameter	Study Group (SG)				Control Group (C)				*p*
	Mean ± OS	Me QD, QG	Min	Max	Mean ± OS	Me QD, QG	Min	Max	
TBW [L]	43.46 ± 11.72	44.60 34.10 48.80	24.60	64.50	37.57 ± 7.26	35.90 32.00 44.55	27.70	51.30	0.161
ECW [L]	19.00 ± 4.67	19.50 15.80 19.80	11.30	27.50	16.13 ± 2.62	15.80 14.25 18.05	12.60	21.10	0.063
ECW/TBW [%]	43.62 ± 3.74	43.40 40.30 45.80	38.80	52.80	43.23 ± 2.31	43.20 41.25 44.60	39.90	49.00	0.912
Resistance	465.13 ± 97.35	476.65 382.50 512.05	313.90	655.70	543.70 ± 74.74	550.40 484.80 604.75	423.20	676.70	0.023
Reactance	48.98 ± 13.04	48.00 36.20 62.25	30.10	67.90	59.88 ± 8.07	60.75 55.05 65.35	41.60	75.30	0.031
VAT [l]	2.93 ± 2.18	2.80 0.80 3.60	0.40	8.20	1.27 ± 1.28	1.00 0.40 1.45	0.10	5.30	0.023
WC [m]	0.98 ± 0.15	1.02 0.86 1.07	0.68	1.25	0.81 ± 0.13	0.84 0.70 0.87	0.66	1.14	0.006
FM [kg]	25.40 ± 10.92	27.25 17.90 29.94	1.95	42.75	18.94 ± 9.76	15.71 13.32 23.24	8.83	50.43	0.028
FMI [kg/m^2^]	8.75 ± 3.59	8.50 7.30 10.00	0.60	14.80	6.65 ± 3.64	5.05 4.50 8.45	3.20	18.10	0.034
FFM [kg]	59.49 ± 16.31	62.05 47.10 67.05	33.06	88.05	50.94 ± 10.09	48.07 43.64 61.09	37.03	69.73	0.146
FFMI [kg/m^2^]	20.25 ± 3.72	20.30 17.80 23.00	14.70	26.30	17.44 ± 2.32	17.15 15.75 18.70	14.20	22.80	0.027

TBW—total body water; ECW—extracellular water; VAT—visceral fat area; WC—waist circumference; FM—fat mass; FMI—fat mass index; FFM—fat-free mass; FFMI—fat-free mass index.

**Table 6 cimb-47-00255-t006:** Pairs of variables for which a statistically significant correlation was demonstrated in the Spearman test in the study and control groups.

Pair of Variables	Spearman’s Correlation Coefficient	*p*
Study Group		
Adiponectin [ng/mL] and insulin [µU/mL]	0.480	0.015
Resistin [ng/mL] and total cholesterol [mg/dL]	0.463	0.020
Control Group		
IL6 [pg/mL] and total cholesterol [mg/dL]	0.440	0.040
IL6 [pg/mL] and triglycerides [mg/dL]	0.493	0.020
IL6 [pg/mL] and non-HDL cholesterol [mg/dL]	0.476	0.025
Leptin [ng/mL] and FM [kg]	0.493	0.027
Resistin [ng/mL] and visfatin [ng/mL]	−0.605	0.003
Visfatin [ng/mL] and FM [kg]	−0.497	0.026

**Table 7 cimb-47-00255-t007:** Values of Spearman’s rank correlation coefficients between the concentrations of individual adipocytokines and parameters describing insulin resistance, lipids, and fat tissue content in the study group.

Parameter	Leptin	*p*-Value	Adiponectin	*p* Value	Resistin	*p* Value	Visfatin	*p*-Value	IL-6	*p*-Value
HOMA-IR	0.13	0.52	−0.12	0.53	−0.21	0.31	0.04	0.82	0.05	0.80
VAT [L]	−0.30	0.32	−0.30	0.32	−0.14	0.63	0.05	0.87	−0.03	0.91
WC [m]	−0.27	0.36	−0.50	0.08	−0.25	0.41	0.11	0.71	−0.08	0.78
FM [kg]	−0.31	0.30	−0.24	0.42	−0.30	0.31	0.36	0.23	−0.04	0.90
FMI [kg/m^2^]	−0.27	0.38	−0.37	0.22	−0.21	0.48	0.39	0.19	0.05	0.86
CH mg/dL	0.03	0.90	0.19	0.36	0.46	0.02	−0.12	0.54	−0.08	0.70
HDL cholesterol mg/dL	0.22	0.29	0.26	0.20	0.37	0.06	0.03	0.90	0.22	0.29
LDL cholesterol mg/dL	−0.16	0.43	0.08	0.70	0.25	0.21	−0.24	0.24	−0.28	0.17
TG mg/dL	−0.16	0.45	−0.14	0.50	0.09	0.65	−0.06	0.75	−0.07	0.72
Adipocytokines										
Leptin [ng/mL]			0.20	0.32	0.12	0.56	0.15	0.46	−0.16	0.42
Adiponectin [ng/mL]	0.20	0.32			0.05	0.80	−0.02	0.92	−0.21	0.31
Resistin [ng/mL]	0.12	0.56	0.05	0.80			−0.27	0.19	0.07	0.72
Visfatin [ng/mL]	0.15	0.46	−0.02	0.92	−0.27	0.19			0.35	0.09
IL-6 [pg/mL]	−0.16	0.42	−0.21	0.31	0.07	0.72	0.35	0.09		

HOMA-IR—homeostatic model of insulin resistance; VAT—visceral fat area; WC—waist circumference; FM—fat mass; FMI—fat mass index; FFM—fat-free mass; FFMI—fat-free mass index; CH—total cholesterol; HDL—high-density lipoprotein cholesterol; LDL—low-density lipoprotein cholesterol; TG—triglycerides.

**Table 8 cimb-47-00255-t008:** Values of Spearman’s rank correlation coefficients between the concentrations of individual adipocytokines and parameters describing insulin resistance, lipids, and fat tissue content in the control group.

Parameter	Leptin	*p*-Value	Adiponectin	*p* Value	Resistin	*p* Value	Visfatin	*p*-Value	IL-6	*p*-Value
HOMA-IR	0.30	0.17	−0.32	0.14	0.09	0.70	−0.02	0.92	−0.04	0.86
VAT [L]	0.11	0.65	0.33	0.15	0.08	0.75	−0.19	0.42	0.10	0.67
WC [m]	0.26	0.27	0.25	0.29	0.05	0.82	−0.23	0.33	0.42	0.07
FM [kg]	0.49	0.02	0.24	0.30	0.12	0.60	−0.50	0.03	0.24	0.31
FMI [kg/m^2^]	0.43	0.06	0.20	0.39	0.02	0.42	−0.43	0.06	0.1	0.66
Cholesterol mg/dL	0.10	0.66	−0.10	0.64	0.02	0.93	−0.15	0.50	0.44	0.04
HDL cholesterol mg/dL	−0.31	0.15	−0.20	0.38	−0.28	0.20	0.19	0.40	−0.23	0.30
LDL cholesterol mg/dL	0.21	0.36	0.19	0.4	0.05	0.81	−0.27	0.22	0.38	0.08
TG mg/dL	0.35	0.11	−0.05	0.40	−0.12	0.60	0.13	0.55	0.49	0.02
Adipocytokines										
Leptin [ng/mL]			−0.05	0.82	−0.107	0.64	−0.045	0.46	0.33	0.14
Adiponectin [ng/mL]	−0.053	0.33			−0.26	0.81	−0.09	0.70	−0.09	0.71
Resistin [ng/mL]	−0.107	0.64	−0.26	0.81			−0.60	0.002	0.12	0.60
Visfatin [ng/mL]	−0.04	0.84	−0.09	0.70	−0.60	0.002			0.06	0.80
IL-6 [pg/mL]	0.33	0.14	−0.09	0.71	0.12	0.60	0.06	0.80		

HOMA-IR—homeostatic model of insulin resistance; VAT—visceral fat area; WC—waist circumference; FM—fat mass; FMI—fat mass index; FFM—fat-free mass; FFMI—fat-free mass index; CH—total cholesterol; HDL—high-density lipoprotein cholesterol; LDL—low-density lipoprotein cholesterol; TG—triglycerides.

## Data Availability

The data presented in this study are available on request from the corresponding author. The data are not publicly due to lack of patients’ consent to making their data public.

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
