# Peer review of "The Effect of Kidney Transplantation and Immunosuppressive Therapy on Adipose Tissue Content and Adipocytokine Plasma Concentration—Preliminary Study"

_cimb, 2025, doi:10.3390/cimb47040255_

Round 1

Reviewer 1 Report

Comments and Suggestions for Authors

Title: The Effect of Kidney Transplantation and Immunosuppressive Therapy on Adipose Tissue Content and Adipocytokine Plasma Concentration-Preliminary Study

Journal: Current Issues in Molecular Biology

Overall assessment:
The manuscript presents a preliminary study on the adipocytokine profile, adipose tissue composition, basic parameters of carbohydrate and lipid metabolism in patients with kidney transplant and immunosuppressive therapy. The authors chose a clinically relevant topic, the involvement of adipokines in kidney transplantation being still a subject of intensive research. The paper is easy to follow and written very well, the introduction provides sufficient background and includes relevant references. For a preliminary work, the methods are suitable and the work design is well described, and the authors presented the limitations of the study. The conclusions are supported by the data.

While the manuscript is scientifically sound, certain areas require improvement to ensure clarity and rigor.

Minor comments:

  1. The figure and table legends should provide more context, they should explain the key takeaways without requiring readers to consult the text.
  2. There are some parts that require modification (table 6 – number of participants; tables 1, 2 and 3 – abbreviation).

Recommendations for Authors:

  1. Please revise figures and tables for better clarity and ensure that legends are sufficiently detailed. I also recommend numbering the tables in order of appearance and adding in the legend the test used to perform the statistical analysis.
  2. Please correct the number of participants in table 6 (study group, n=23 instead of 25) and the abbreviation for SD, LQ and UQ (Mean ±OS, Me QD, QG) in tables 1, 2 and 3.

Decision: Consider after Minor Revision

Author Response

 Thank you for the professional review

  1. The figure and table legends have been completed  
  2. We have made the suggested changes in point 2 (table 6 – number of participants; tables 1, 2 and 3 – abbreviation).

Reviewer 2 Report

Comments and Suggestions for Authors

The  manuscript entitled “The Effect of Kidney Transplantation and Immunosuppressive Therapy on Adipose Tissue Content and Adipocytokine 3 Plasma Concentration-Preliminary Study” is informative and I have some comments for the manuscript improvement. My comments are as follows

  1. The abstract part seems to be length and I recommend the authors to reduce the content of the abstract.
  2. The figure1 picture clarity can be improved
  3. I suggest the authors add the novelty of the research in the introduction section.  
  4. I recommend the authors to mention the table data in a better way. Some of the numbers mentioned in the table seem to be overlapping.
  5.  
  6.  Line number-432, The following sentences start with “In the conducted study, increase in the …….. abdominal fat tissue” need revisions.
  7. Line number-472, The sentences start with “In order to eliminate potential measurement errors, patients were … has some linguistic issues. Hence, the sentences need to be rephrased 

Author Response

Thank you for the revision

  1. The abstract has been shortened
  2. The description under Fig 1 has been expanded with additional information
  3.  The introduction section takes into account new data.  
  4.  The table data has been corrected.
  5. The data in the tables has been corrected
  6.  Line number-432, The sentence has been corrected.
  7.  The sentences in line number -472 has been  rephrased 

Reviewer 3 Report

Comments and Suggestions for Authors

The manuscript aimed to evaluate the changes in adipocytokine profiles, interleukin-6 (IL-6) levels, and body composition after kidney transplantation. The authors concluded that kidney transplantation leads to significant alterations in adipocytokines levels, with potential implications for metabolic health, however, the results are based only on a single observation. In my opinion, the examinations should be performed at defined time-points after the transplantation to give an insight of the changes among renal transplant recipients. As there were differences in age and body weight between study and control group, the results may not reflect the status of renal transplant recipients. Renal transplant recipients had higher body mass what might have influenced the results. The control group should match the study group in the characteristics. My other comments are listed below:

  • The abbreviations should be checked throughout the study. In the Introduction, the abbreviations TAC, MMF and GCS are used, but they are explained only in Materials and Methods.
  • The study group and the control group differ in age and body weight. In my opinion, it may influence the results to some extent. How was the study group selected?
  • I suggest adding the information about the time that has passed since the transplant. Was it wide range or an early post-transplant time?
  • The details on assays detection range, sensitivity, and precision should be presented in a table as they do not form a correct sentence (lines 180-186).
  • Line 212: HOMA index should be explained what it consists of and means.
  • Figure 2: why were the letters ‘B’ and ‘K’ chosen for control and study group? It is difficult to follow the results with these letters that do not match control and study group naturally.
  • Table 5 does not include p values so the data are not fully presented.
  • The Discussion is long and on the first page most of the Discussion focuses on the literature data, while the results of the current study are not fully explained and discussed. I suggest rewriting the Discussion and discussing the results along with the literature data.
  • Lines 350, 384: no references to the literature.

Author Response

Thank you for the revision

1.The abbreviations has been corrected

2. The groups differed in terms of age because the assumption was to examine healthy volunteer, which also translates into their younger age and body weight

3. The period that has passed since the transplantation was taken into account

4.The details on assays detection range, sensitivity, and precision has been presented in a table. 

5. HOMA index now is explained what it consists of and means.

6. Letters ‘B’ and ‘K’ chosen for control and study group has been changed

7. Table 5 has been withdrawn. The data  has been  entered into two tables taking into account p value.

8. The Discussion has been corrected. 

9. Lines 350, 384- has been removed

Reviewer 4 Report

Comments and Suggestions for Authors

The manuscript entitled "The effect of kidney transplantation and immunosuppressive therapy on adipose tissue content and adipocytokine plasma concentration - preliminary study" addresses an existing clinical problem, namely the effects of immunosuppressive therapy after kidney transplantation (KTx) on adipose tissue distribution and adipokine levels. Although these factors have been implicated in metabolic diseases and long-term outcomes in transplant recipients, the subject is not novel and this preliminary study provides some insight into this topic by adding to the existing literature. However, the authors should address the concerns raised by this reviewer, and clarify and expand on the following issues:

  1. The most important limitation of the study design is that the effects of kidney transplantation is measured along with the effects of immunesuppressive therapy. The authors did their best to interindividual variability by applying thoughtful inclusion/exclusion criteria (e.g.,  uniform immunosuppressive regimen: tacrolimus + mycophenolate mofetil + glucocorticoids), but the generalizability of the results is very low. However, the authors should discuss whether similar patterns have been observed in other solid-organ transplant recipients (e.g., liver or heart transplant) could offer additional perspective on whether the observed adipocytokine shifts are unique to kidney transplantation or reflect a more general response to immunosuppression.
  2. The second important issue is the effects of glucocorticoids on adipose tissue metabolism and adipokines. This has widely been studied previously, and by discussing the effects of glucocorticoids only, would enhance the interpretability of the results.
  3. The next important issue is the calculation of effect sizes and the number of subjects. Although the authors are aware that this is a preliminary study with a small number of subject, they should present a priori calculations how the minimum number of subjects was determined. This is particularly important for understanding relationships that may become clearer or more nuanced with a larger cohort (e.g., changes in visfatin or resistin).
  4. Although the article outlines potential pathways by which immunosuppressive therapy (especially glucocorticoids) can alter adipose tissue functions, further elaboration on mechanistic pathways would enrich the discussion. For instance, discussing how tacrolimus or steroids might modulate leptin signaling or adiponectin expression at the molecular level would give deeper insight.

Minor obs.:

Words adipokines and adipocytokines are used alternatively. They are synomyms but please be consequent throuthout the manuscript.

Abstract:

Line 22 - glucocorticoids

Line 23,24 - abbreviations should be avoided in the Abstract

Line 25 - equipment should only be mentioned in the Materials and methods.

 Introduction:

Too long, it should be compacted.

Materials and methods:

Manufacturer should be provided for Seca mBCA 525 medical body composition analyzer.

Results

Rezistin or resistin - please be correct.

Discussion

In general, the discussion is too long. It should be compacted.

Line 336-341 - this paragpraph needs references

Comments on the Quality of English Language

The English could be improved to more clearly express the research.

Author Response

Thank you for the review. We agree with these comments,
1. the discussion was extended to include the results of similar studies obtained in people after solid organ transplantation. For patients who have undergone solid organ transplantation, immunosuppressive therapy is always used. Since the effects of transplantation (acute rejection episodes, infections, adverse drug reactions ) depend on immunosuppressive therapy, it is natural that these data are analyzed together. Only in animal studies is it possible to assess the effects of transplantation without the use of immunosuppressive drugs.

2. The effects of glucocorticoids have been more extensively described

3.We acknowledge that the relatively small number of participants (n=47) may be considered a
limitation. Therefore, we performed an a priori sample size calculation during the study
design phase to estimate the minimum number of subjects required to detect significant
differences between groups with adequate statistical power.
The sample size was calculated based on an expected difference in mean leptin concentrations
between kidney transplant recipients and healthy controls. Drawing on previously published
data (e.g., Fonseca et al., 2015; Sukackiene et al., 2021), we assumed a mean difference of
approximately 8 ng/mL with a standard deviation of 10 ng/mL. Using a significance level of α
= 0.05 and a power of 1–β = 0.8 (80%), the calculation (performed for both the t-test for
independent samples and the non-parametric Mann–Whitney U test, considering possible
deviations from normality) indicated that a minimum of 21 participants per group was
required.
Our final study included 25 kidney transplant recipients and 22 control participants, which
meets the threshold suggested by the power analysis for detecting group differences in the
primary outcome — serum leptin concentration. Additionally, all statistical analyses were
conducted using non-parametric methods (Mann–Whitney U test and Spearman’s rank
correlation) to account for data distribution and sample size.
It is important to note that this study was designed as a preliminary pilot investigation.
Therefore, strict inclusion criteria were applied (e.g., uniform immunosuppressive regimen
with TAC + MMF + GCS, stable graft function, no diabetes, and no prior rejection episodes)
to reduce confounding variables and group heterogeneity. This methodological rigor helped
ensure data quality despite the moderate sample size.

4. To the best of our knowledge, the metabolic pathways of immunosuppressive drugs are different from those of adipocytokines. Therefore, there is no data available in the literature regarding their mutual relationship

Round 2

Reviewer 3 Report

Comments and Suggestions for Authors

All the remarks except for the aim of the study have been addressed.

Author Response

Thank you for the review. We described the aim of the study in detail.

Reviewer 4 Report

Comments and Suggestions for Authors

The authors responded partially to the concerns raised by the reviewer. Consequently, some improvement of the quality of the manuscript can be observed, but the initially proposed refinements should be implemented.

1. The reviewer is fully aware that kidney transplanted patients cannot be leaved without immunosuppressive treatment. However, as it was highlighted, the manuscript could provide some details on this issue by patients with different transplanted solid organs who also require immunosuppression (if there is data available).

2. This comment has been addressed partially.

3. This comment has been addressed adequately.

4. The authors misinterpreted the reviewer's suggestion. As proposed, discussing how tacrolimus or glucocorticoids might modulate leptin signaling or adiponectin expression at the molecular level would give deeper insight.

This referred to the direct inhibition of adiponectin signaling by glucocorticoids, which was broadly studied and discussed previously. For a good summary of the existing literature see: 10.1016/B978-0-12-398313-8.00007-5

Although it is not completely understood how GC decrease adiponectin levels in KTx patients, the issue should be discussed in conjunction with insulin and plasma glucose levels, which were significantly elevated in the study group. 

Comments on the Quality of English Language

The manuscript would benefit from a thorough proofreading.

Author Response

Comment 1. Thank you for the review. Information on the role of adipocytokines in patients after solid organ transplantation has been included, but there are no broader studies on this issue in the scientific literature

Comment 4. We supplemented the data according to the reviewer's suggestion with information from the recommended publication.10.1016/B978-0-12-398313-8.00007-5. 

In the available literature, there was only one study in patients after kidney transplantation, and only correlations were studied. Therefore, the mechanism how GC decrease adiponectin levels is probably not yet known.